# *Choiromyces sichuanensis* sp. nov., a New Species from Southwest China, and Its Mycorrhizal Synthesis with Three Native Conifers

**Ran Wang** [1,2,3], **Shanping Wan** [4,*], **Juan Yang** [1] **and Fuqiang Yu** [1,2,*]

1   Germplasm Bank of Wild Species, Kunming Institute of Botany, Chinese Academy of Sciences, 132 Lanhei Road, Kunming 650201, China
2   Yunnan Key Laboratory for Fungal Diversity and Green Development, Kunming Institute of Botany, Chinese Academy of Sciences, 132 Lanhei Road, Kunming 650201, China
3   Department of Crop and Forest Science, University of Lleida, Av. Alcalde Rovira Roure, 191, 25198 Lleida, Spain
4   College of Resource and Environment, Yunnan Agricultural University, Kunming 650201, China
*   Correspondence: wsp871117@163.com (S.W.); fqyu@mail.kib.ac.cn (F.Y.)

**Abstract:** A new *Choiromyces* species was discovered at local wild mushroom markets in Songpan County, Sichuan, southwest China where it has been considered as a Chinese white truffle. Based on both morphological and phylogenetic analyses, the collection was described as *Choiromyces sichuanensis* sp. nov. This study confirms the occurrence of members of *Choiromyces* in China. In addition, the mycorrhizal synthesis via spore inoculation between *C. sichuanensis* and *Pinus armandii* or two *Picea* species of *Pi. likiangensis* and *Pi. crassifolia* was attempted in a greenhouse. Both morphoanatomical and molecular analyses evidenced well-developed mycorrhization between *C. sichuanensis* and *P. armandii*, but not in *Picea* seedlings. Our current study provides data about the species diversity and mycorrhizal research of this genus for further studies. In addition, a successful mycorrhization between *C. sichuanensis* and selected tree species, irrespective of Pinus genus or other plant species, would broaden the set of species for a successful mycorrhization in greenhouse conditions and potential outplanting for cultivation purposes.

**Keywords:** ectomycorrhizae; pig truffle; hypogeous fungi; morphology; phylogeny

## 1. Introduction

The genus *Choiromyces* Vittad. (Tuberaceae, Pezizales, Ascomycotina) with *C. meandriformis* Vittad. as the type species was first described in 1831 [1]. Although *Choiromyces* is widely distributed in Asia, Europe, and North America [2–5], the species diversity of *Choiromyces* is low and only eight species have been categorized (*C. alveolatus*, *C. cerebriformis*, *C. cookei*, *C. ellipsosporus*, *C. helanshanensis*, *C. meandriformis*, *C. tetrasporus* and *C. venosus*) [2–7]. Among them, *C. venosus* is commonly considered conspecific with *C. meandriformis* [4,8], while *C. cerebriformis* and *C. helanshanensis* are sporadically reported from China [4,5].

*Choiromyces venosus* has been a delicacy in Northern Europe [9]. It is hypogeous, subglobose, or irregular, whitish in color, has a fragrant ascomata, and is solid gleba, which is often mistaken for a species of the genus *Tuber*. For example, *C. meandriformis* was once sold as *T. magnatum* Picco. as a highly prized mushroom in the European markets [3]. Interestingly, a similar situation also occurred in 2020 in the local fresh mushroom markets of Sonpan county, Sichuan, southwest China where "white truffle-like" mushrooms were sold as Chinese white truffles (*Tuber panzhihuanense* Deng, X.J. & Wang, Y.) at a comparatively high price of $80−100 USD kg$^{-1}$. Our morphological and molecular phylogenetic analyses showed that these market collections belonged to a new species that had not been previously classified.

By enquiring local collectors and identifying plant leaves on the fruiting bodies, we assumed that those mushroom samples grow under a *Picea* dominated forest at altitude of ~2000 m. *Choiromyces* species has been described to be able to establish ectomycorrhizal associations in the field with Rosaceae (*Dryas*), Salicaceae (*Salix*), and Pinaceae (*Abies*, *Picea*, *Pinus*) and thus play an important ecological role in these forest ecosystems [4,5,10]. However, no information is available on how such an ectomycorrhization could be constructed under laboratory environments. In this study, using spores of *C. sichuanensis* sp. nov., we artificially examined their mycorrhizal symbiosis of three selected tree species including *Pinus armandii* Franch, an economically planted tree species that is also widely and naturally distributed at high altitude in southwest China [11] and two other important afforestation tree species, *Picea likiangensis* (Franch.) Pritz. and *Pi. crassifolia* Kom., which have wide distributions in southwest China and northwest China, respectively. Herein, we described these collections as *Choiromyces sichuanensis* sp. nov., and its relationships with other *Choiromyces* species were discussed. Additionally, a successful mycorrhizal synthesis between *C. sichuanensis* and selected tree species, irrespective of *Pinus* genus or other plant species, would broaden the set of species for a successful mycorrhization in greenhouse conditions and develop an effective pathway for cultivation purposes.

## 2. Materials and Methods

### 2.1. Morphological Studies of Ascomata

Fresh specimens were purchased from markets in Songpan County, Tibetan Qiang Autonomous Prefecture of Ngawa, Sichuan, China. Morphological examinations were performed for three *Choiromyces* specimens, comprising 14 ascomata, which are from three different collection sites. Gross morphology was described based on the fresh ascomata, and microscopic examination was later conducted using dry material according to Yang and Zhang [12]. Hand-cut sections were mounted in a 5% (*w/v*) KOH solution and examined under a light microscope (Leica DM2500, Leica Microsystems, Wetzlar, Germany) to observe and record morphology of the ascospores, dermatocystidia, and peridium. For the evaluation of the range of ascus and spore size, 30 asci and 80 ascospores were totally measured based on three samples from three different sites. The spores were also scanned by electron microscopy to observe the surface decoration and record spines' length. For the scanning electron microscopy (SEM) observation, spores were scraped from the dried gleba onto double-sided tape and then directly mounted on a SEM stub, coated with gold-palladium, examined, and photographed using a JSM-5600LV SEM (JEOL, Tokyo, Japan). Specimens were deposited at Yunnan Agricultural University, Kunming, Yunnan, China.

### 2.2. Mycorrhizal Synthesis with Plants

Seeds of *Pinus armandii* were purchased from the Ciba market, Kunming, Yunnan. Seeds of *Picea likiangensis* (No. 16CS14028) and *Pi. crassifolia* (No. LiuJQ-09XZ-LZT-125) were obtained from the Southwest Germplasm Bank of Wild Species, Kunming Institute of Botany, Chinese Academy of Sciences. In June 2021, 30% hydrogen peroxide was used for surface sterilization of seeds for 10 min and then rinsed thoroughly in distilled water. Seeds of all three species were germinated on a sterilized mixture of perlite and peat (1:1, V:V), which was autoclaved at 121 °C for one hour. Substrate for inoculation was composed of vermiculite: peat: perlite (3:2:1, V:V:V) [13]. Before inoculation, one ascomata was sliced, air dried (average room temperature was 12 °C), and stored at −40 °C. Three-month-old seedlings (September 2021) were rinsed with tap water to remove remaining substrate, and roots were kept moistened until inoculation. Dry slices of ascomata were soaked in distilled water for 24 h at 4 °C and then homogenized with a blender. Spore concentration was measured with a hemacytometer. Five seedlings of each tree species were inoculated with *C. sichuanensis* spore slurry. Each seedling received $8.0 \times 10^6$ spores. Spore slurry was applied in the upper third of the seedling's root system. Five seedlings of each non-*C. sichuanensis* inoculated control tree species received 10 mL of distilled water. Seedlings were kept in 420 mL square plastic containers ("olive pots", Daltons Ltd., Christchurch,

New Zealand) and grown in a greenhouse on the Kunming Institute of Botany (KIB) campus for six months (September 2021 to March 2022). Growth conditions were natural day-length (10–12 h), average temperature in the day was 14–22 °C. The containers with or without *C. sichuanensis* inoculation were grown in groups, rather than a randomized arrangement. Each group consisted of a given *Pinus/Picea* species inoculated or not with *C. sichuanensis*: 10 seedlings distributed in two rows (one inoculated, one uninoculated) of five seedlings. Seedlings were watered with tap water three times a week to maintain moisture around 50–80%.

### 2.3. Morphological Observations of Ectomycorrhizae (ECM)

Each seedling was carefully removed from the container to identify the macro-morp hological and anatomical characters of *C. sichuanensis* ECM using a stereomicroscope (Leica S8AP0, Leica Microsytems, Wetzlar, Germany) and a compound light microscope (Leica DM2500, Leica Microsytems, Wetzlar, Germany) following the methods of Agerer [14] and Munsell Color Chart (1994 revised edition). Photographs were captured using the Leica Application Suite. Cross- and longitudinal-sections were made by a freezing microtome (Leica CM3050S) and observed under a compound light microscope (Leica DM2500) for characterization [15,16]. ECM colonization was calculated as the number of ECM roots divided by the total number of root tips, which include ECM root tips and non-ECM root tips [17]. Number of root tips were estimated using $W_{IN}R_{HIZO}$ (Regent Instruments Canada Inc., Quebec, QC, Canada).

### 2.4. Phylogenetic Analyses

Total DNA was extracted from pieces of dried ascomata using a modified CTAB procedure [18]. Genomic DNA of 10 pooled mycorrhizal tips displaying morphology characteristic of *C. sichuanensis* from each inoculated seedling and 10 pooled root tips from each non-inoculated seedling were extracted using Aidlab™ kit (Beijing, China). The ITS region of nuclear ribosomal DNA (nrDNA) was amplified using primers ITS1F and ITS4 [18,19]. Polymerase chain reactions (PCR) were performed using the following procedure: 25 μL of PCR reaction solution contained 1 μL DNA, 1 μL (5 μM) of each primer, 2.5 μL 10 × buffer ($Mg^{2+}$ dNTP (1 mM), 0.5 μL BSA (0.1%), 0.5 μL $MgCl_2$, 1 U of Taq DNA polymerase (Takara Taq, Takara Biotechnology, Dalian, China). PCR reactions were run as follows: 94 °C for 5 min, followed by 35 cycles of 94 °C for 30 s, 52 °C for 1 min, and 72 °C for 1 min, followed by a final extension at 72 °C for 10 min. The PCR products were sent to Tsingke Biotechnology Co., Ltd. (Beijing, China) for sequencing.

We have obtained three sequences from ascocarps, 1 from ectomycorrhiza, and the rest of the 52 sequences used for phylogenetic analysis (Table 1) were retrieved from the GenBank (https://www.ncbi.nlm.nih.gov/genbank/). *Dingleya* sp. (JF300131, HM485334, JQ925627 and JQ925628) were selected as the outgroup (Table 1) [4,5]. Dataset was aligned using MAFFT v.7.0 [20] applying the L–INS–I strategy and then manually edited with BioEdit v.7.0.9 as needed [21] for deleting the bases in the beginning and end of the matrix that do not belong to the ITS interval. The phylogenetic relationships of taxa were inferred using maximum likelihood (ML) [22] and Bayesian inference (BI) [23].

**Table 1.** Species name, voucher, origin, ITS, and references for sequences.

| Species Name | Voucher | Origin | ITS | References |
|---|---|---|---|---|
| *Choiromyces alveolatus* (Harkn.) Trappe | MES97 | USA | HM485332 | Bonito et al. [24] |
| *C. alveolatus* | Trappe 17497 | USA | EU669384 | Unpublished |
| *C. alveolatus* | 22830 | Israel | AF501258 | Ferdman et al. [25] |
| *C. alveolatus* | HS2886 | USA | HM485333 | Bonito et al. [24] |
| *C. cerebriformis* Yuan, T.J., Li, S.H. & Wang, Y. | YAAS 8890 Holotype | Diqing, China | MW209701 | Yuan et al. [5] |

**Table 1.** *Cont.*

| Species Name | Voucher | Origin | ITS | References |
|---|---|---|---|---|
| *C. cerebriformis* | YAAS TJ16-2 | Yunnan, China | MT672014 | Yuan et al. [5] |
| *C. cerebriformis* | YAAS TJ16-1 | Yunnan, China | MT672013 | Yuan et al. [5] |
| *C. helanshanensis* Juan Chen & P.G. Liu | YAAS L3063 | China | MT672012 | Yuan et al. [5] |
| *C. helanshanensis* | YAAS L3062 | China | MT672011 | Yuan et al. [5] |
| *C. helanshanensis* | YAAS L3051 | China | MT672010 | Yuan et al. [5] |
| *C. helanshanensis* | HKAS 80639 | Inner Mongolia, China | KP019351 | Chen et al. [4] |
| *C. helanshanensis* | HKAS 80647 | Inner Mongolia, China | KP019350 | Chen et al. [4] |
| *C. helanshanensis* | HKAS80631 | Inner Mongolia, China | KP019349 | Chen et al. [4] |
| *C. helanshanensis* | HKAS80642 | Inner Mongolia, China | KP019348 | Chen et al. [4] |
| *C. helanshanensis* | HKAS80646 | Inner Mongolia, China | KP019347 | Chen et al. [4] |
| *C. helanshanensis* | HKAS80634 Holotype | Inner Mongolia, China | KP019346 | Chen et al. [4] |
| *C. helanshanensis* | HKAS80645 | Inner Mongolia, China | KP019345 | Chen et al. [4] |
| *C. helanshanensis* | HKAS80638 | Inner Mongolia, China | KP019344 | Chen et al. [4] |
| *C. helanshanensis* | HKAS80636 | Inner Mongolia, China | KP019343 | Chen et al. [4] |
| *C. helanshanensis* | HMAS83766 | Heilongjiang, China | KU531609 | Unpublished |
| *C. helanshanensis* | HKAS80641 | Inner Mongolia, China | KU531606 | Unpublished |
| *C. magnusii* (Mattir.) Paol. | AH11894 | Spain | JF300144 | Moreno et al. [3] |
| *C. magnusii* | AH19770 | Spain | JF300143 | Moreno et al. [3] |
| *C. meandriformis* Vittad. | K(M):171388 | United Kingdom | MZ159438 | Unpublished |
| *C. meandriformis* | K(M)53644 | England | EU784185 | Brock et al. [26] |
| *C. meandriformis* | K(M)135393 | England | EU784184 | Brock et al. [26] |
| **C. sichuanensis** Wan, S.P., Wang, R. & Yu, F.Q. | **YNAU003 Holotype** | **Sichuan, China** | **MW380902** | **This study** |
| *C. sichuanensis* | **YNAU004** | **Sichuan, China** | **OK585070** | **This study** |
| *C. sichuanensis* | **YNAU0022** | **Sichuan, China** | **OM417587** | **This study** |
| **C. sichuanensis** Wan, S.P., Wang, R. & Yu, F.Q. + **Pinus armandii** Franch | **Synthesized ECM of** *Pinus armandii* | **Sichuan, China** | **ON113253** | **This study** |
| *C. venosus* (Fr.) Th. Fr. | AH38904 | Romania | JF300147 | Moreno et al. [3] |
| *C. venosus* | AH38915 | Italy | JF300146 | Moreno et al. [3] |
| *C. venosus* | AH38935 | UK | JF300145 | Moreno et al. [3] |
| *Choiromyces* sp. | AR1291 | Canada | JX630965 | Timling et al. [27] |
| *Choiromyces* sp. | AR1293 | Canada | JX630960 | Timling et al. [27] |
| *Choiromyces* sp. | AR1647 | Canada | JX630948 | Timling et al. [27] |
| *Choiromyces* sp. | HV_D3_8 | USA | JX630713 | Timling et al. [27] |
| *Choiromyces* sp. | PP_S3_3_287_1 | Canada | JX630610 | Timling et al. [27] |
| *Choiromyces* sp. | FB_D3_5_188_1 | USA | JX630491 | Timling et al. [27] |
| *Choiromyces* sp. | FB_D3_1_184a_4 | USA | JX630489 | Timling et al. [27] |
| *Choiromyces* sp. | JLF1765 | USA | MH220314 | Unpublished |
| *Choiromyces* sp. | 58-351 | USA | MH038084 | Hart et al. [28] |
| Uncultured *Choiromyces* | NX1-4299 | Ningxia, China | LC622741 | Unpublished |
| Uncultured *Choiromyces* | NX2-148 | Ningxia, China | LC623368 | Unpublished |
| Uncultured *Choiromyces* | t24 | Xinjiang, China | MF142389 | Unpublished |
| Uncultured *Choiromyces* | NX1-4464 | Ningxia, China | LC622777 | Unpublished |
| Uncultured *Choiromyces* | NX1-4377 | Ningxia, China | LC622757 | Unpublished |
| Uncultured *Choiromyces* | NX1-4472 | Ningxia, China | LC622778 | Unpublished |
| Uncultured *Choiromyces* | NX2-137 | Ningxia, China | LC623365 | Unpublished |
| Uncultured *Choiromyces* | 3232A6 | USA | KF617408 | Taylor et al. [29] |
| Uncultured *Choiromyces* | AR699 | Canada | JX630936 | Timling et al. [27] |
| Uncultured *Choiromyces* | TK21_OTU183 | USA | EF434127 | Taylor et al. [29] |
| *Dingleya* sp. | JT31036 | Australia | JQ925628 | Bonito et al. [30] |
| *Dingleya* sp. | JT31575 | Australia | HM485334 | Bonito et al. [24] |
| *Dingleya* sp. | JT27686 | Australia | JQ925627 | Bonito et al. [30] |
| *Dingleya* sp. | AH37860 | Australia | JF300131 | Moreno et al. [3] |

## 3. Results

### 3.1. Taxonomy

***Choiromyces sichuanensis*** Wan, S.P., Wang, R. & Yu, F.Q., **sp. nov.** (Figure 1).

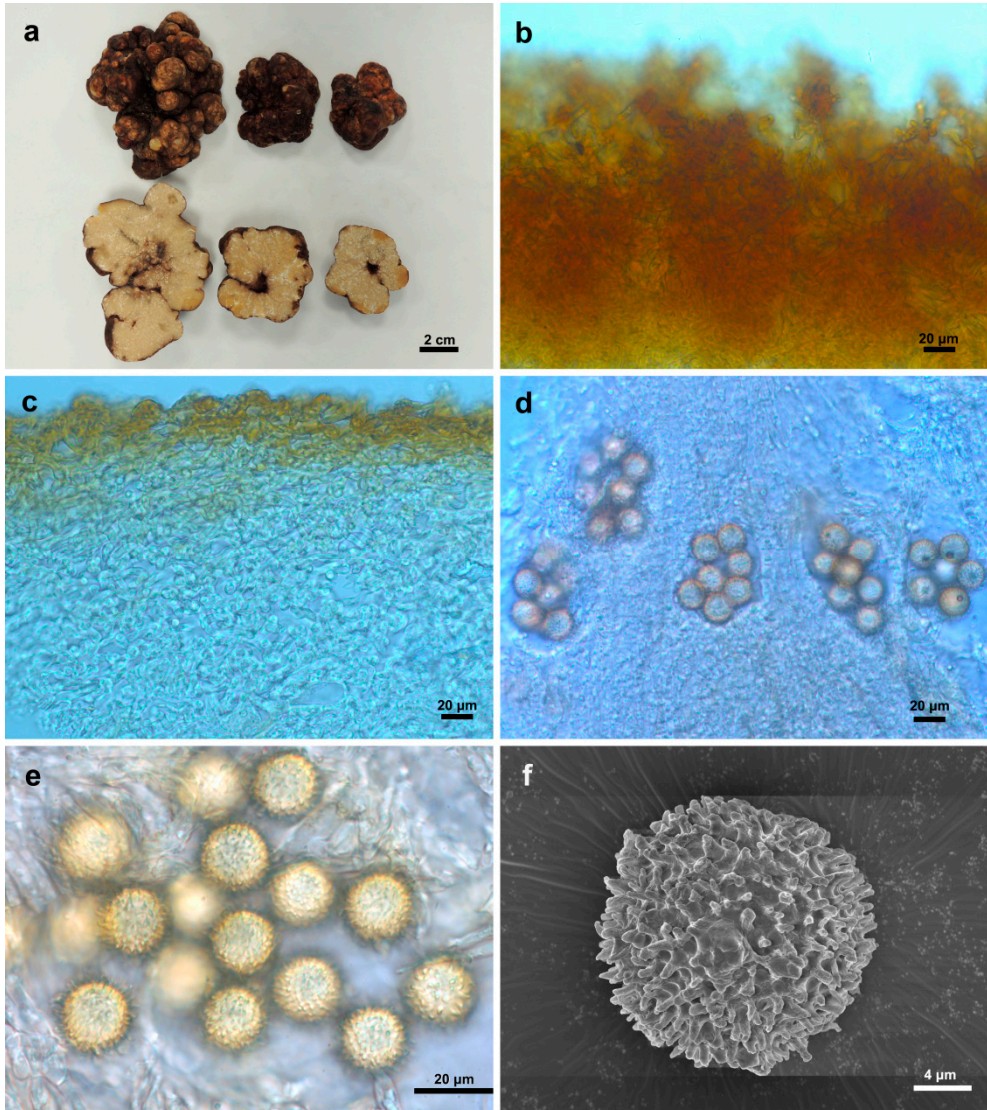

**Figure 1.** *Choiromyces sichuanensis* (YNAU003, Type): (**a**) ascoma; (**b**) dermatocystidia; (**c**) peridium section; (**d**,**e**) light micrograph (LM) of ascospores; (**f**) SEM image of an ascospore.

MycoBank no.: MB843523

**Type**: CHINA, Sichuan Province, in humic soil under *Picea asperata* M ast. forest, 29 October 2020, wsp971 (Holotype, YNAU003, GenBank Acc. No.: ITS = MW380902).

**Etymology**: sichuanensis, referring to the location of the type collection.

Description: The ascomata was 2–9 cm in diam, hypogeous, subglobose, or tuberiform, knobby with deep lobes, surface smooth, white, pale yellow to reddish-brown with dark gullies when fresh (Figure 1a), then becoming brown when dry. The peridium was 37–567 μm thick, not clearly differentiated, composed of light brown (outer) or hyaline (inner) hyphae, 0.7–10.0 μm at septa in diam (Figure 1b,c). The gleba was solid, milky white, slightly brown and marbled with white veins, of interwoven thin-walled hyphae, and an odor that was strong, pleasant. The asci hyaline, clavate to saccate, were eight-spored, (35.0–)50.0–88.0(–103.5) × (31.0–)34.0–65.0(–67.0) μm (*N* = 30). The ascospores were spherical, pale yellow-brown at maturity, (16.7–)17.0–21.4(–24.6) μm in diam (*N* = 80), irregularly covered with curved blunt spines up to 4 μm (Figure 1d–f).

Habitat and distribution: The specimens were found and collected from a humus soil under a *Picea* sp. forest in October. Additional specimens examined: CHINA, Sichuan Province: *Picea* sp., 29 October 2020, wsp 971-1 (YNAU0022, GenBank Acc. No.: ITS = OM417587); wsp 971-2 (YNAU004, GenBank Acc. No.: ITS = OK585070).

Remarks: *Choiromyces sichuanensis* is characterized by hypogeous, glabrous, white, pale yellow or brown, irregularly knobby, much lobed ascomata, and globose spores ornamented with conspicuous, straight, or slightly curved blunt spines. In fact, *C. sichuanensis* can easily be distinguished from the other two phylogenetically related Chinese *Choiromyces* species based on the ascosporal ornamentation with long blunt spines (up to 4 μm) whereas spines' height was <3 μm in Chinese *C. helanshanensis* (2.0–2.5 μm), *C. cerebriformis* (2.0–3.0 μm), and southern African *C. echinulatus* (<2 μm) [4,5,25]. In addition, the spores of *C. meandriformis* are ornamented with free and "hollow" rods with truncated tips (with apical depressions) [3]. *Choiromyces venosus* is commonly considered conspecific with *C. meandriformis* [4]. Additionally, spores' ornamentation of *C. magnusii* looks pitted to verrucose-reticulated with a minutely meshed reticulum, and spores of the American species of *C. alveolatus* are ornamented small pores, which are clearly different from those of *C. sichuanensis* [3]. Additionally, this species shares similar characteristics with *Tuber panzhihuanense*, namely the size of ascocarps, color, an irregularly knobbed ascomata, and whitish gleba with sinuous veins when young. However, *C. sichuanensis* differs from *T. panzhihuanense* and any other white *Tuber* species in its globose spores ornamented with conspicuous spines.

### 3.2. Description of Ectomycorrhizae Choiromyces sichuanensis Wan, S.P., Wang, R. & Yu, F.Q. + Pinus armandii Franch (Figure 2)

After inoculation in the greenhouse for six months (September 2021 to March 2022), ECM was formed between *C. sichuanensis* and all inoculated *P. armandii* seedlings (Figure 2a–c). ECM colonization (%) was 82.86 (±2.04). No ECM was detected in any of *Picea* or control seedlings.

ECM systems were 2.5–4.3 mm long and 1.5–5.6 mm wide (Figure 2b,c), monopodial or dichotomously branched and complex, with a coralloid, main axis 0.3–0.4 mm wide, branches 2.4–6.0 mm long and 0.3–0.5 mm wide, straight to slightly sinuous, and yellowish-brown ochre. Mantles were smooth or slightly woolly (Figure 2d); emanating hyphae were long, with sample or branched, septate, rounded tips, up to 3.0–4.8 μm diam at the base and were frequent on young ECM, rhizomorphs occurrence (Figure 2e). Whole mycorrhiza discoloring was darker with age; the was mantle distinct, not transparent; cortical cells were not visible.

*Cross-section*: The mantle was 18–45 μm thick, with 4–9 layers of hypha cells (Figure 2g,h). The outer mantle layer (Figure 2j) was pseudoparenchymatous or a transitional type between plectenchymatous to pseudoparenchymatous in plan views (close to type H), partially Hartig net-lik with some of them cut in a cross-section view, thin-walled, and smooth. The inner mantle layer (Figure 2f) was plectenchymatous with a net-like arrangement of hyphal bundles (mantle type A), partially Hartig net-like with often thin-walled. Hartig net palmetto type had single hyphal rows (Figure 2h). *Longitudinal section*: The mantle was disposed in a plectenchymatous arrangement; hyphae was disposed in parallel and was flat-lying, compact in the inner layer, compact in the outer mantle, with tannin cells cylindrical, cortical cells rectangular to rounded, and hyphae between cortical cells cylindrical (Figure 2i).

The ECM of other fungal species were not detected in any of the inoculated or control seedlings.

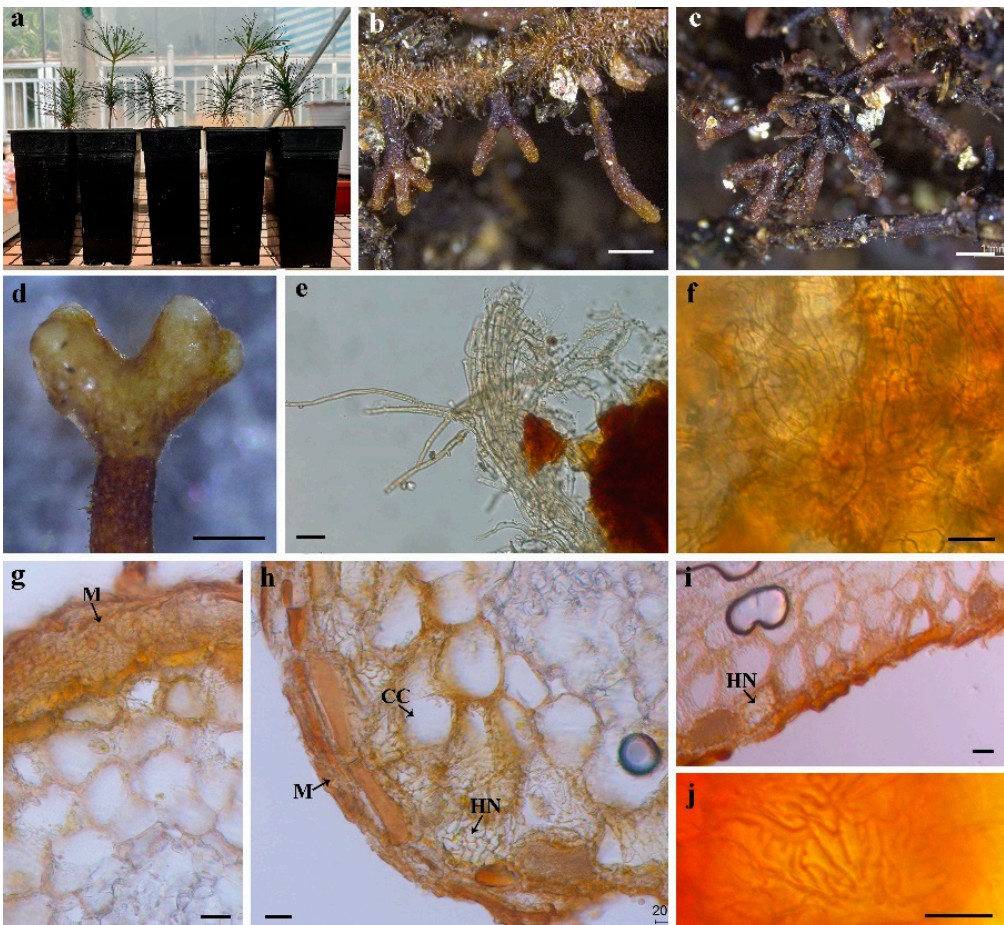

**Figure 2.** (**a**) Six-month-old *Pinus armandii* seedlings inoculated with *Choiromyces sichuanensis* in pots. (**b**,**c**) Ectomycorrhizal tips after six-month inoculation (*bars* = 1 mm). (**d**) Detail of the surface showing sparse emanating hyphae (*bars* = 1 mm). (**e**) Septate and see-through hyphae emanating and rhizomorphs from the outer mantle layer (*bar* = 20 μm). (**f**) Plan view of the inner mantle (*bar* = 20 μm). (**g**,**h**) Cross-section showing the mantle (M), cortical cells (CC), and Hartig net (HN) (*bar* = 20 μm). (**i**) Tangential section showing Hartig net (HN) (*bar* = 20 μm). (**j**) Plan view of the outer mantle showing irregular tree branch–like pattern (*bar* = 10 μm).

*3.3. Phylogenetic Analyses*

The final ITS alignment included 56 sequences (Table 1), which contained 665 aligned sites. The Bayesian analysis yielded similar trees as the parsimony analysis; thus, only the phylogenetic tree established from the parsimony analysis was presented (Figure 3). All analyzed *Choiromyces* species formed a monophyletic group with 100% bootstrap support. Three *Choiromyces* specimens and one *Choiromyces* mycorrhiza from this study were independently clustered as a well-supported subclade (PP = 1.0, BP = 100) and were clearly distinct from the known *C. helanshanensis* and *C. cerebriformis*.

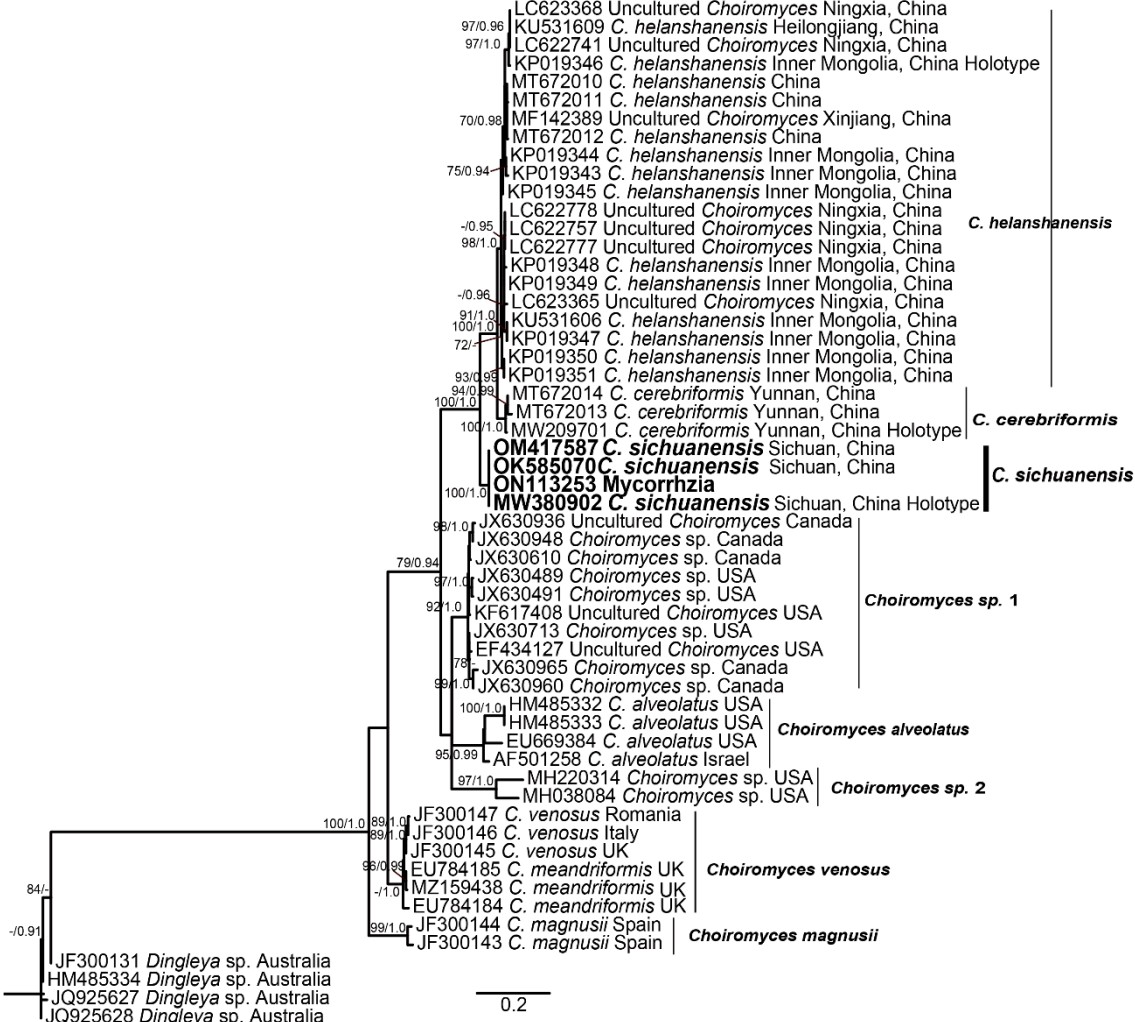

**Figure 3.** RAxML tree based on ITS sequences of *Choiromyces sichuanensis* and related species. Bootstrap (BS) values derived from maximum likelihood (ML) analysis (≥70%) and posterior probabilities (PPs) from Bayesian inference (≥0.90) are shown above or beneath the branches at nodes. New sequences are in colored bold font.

## 4. Discussion

Phylogenetic analysis based on ITS sequences indicated that there are nine species of *C. cerebriformis*, *C. helanshanensis*, *C. sichuanensis*, *Choiromyces* sp. 1, *C. alveolatus*, *Choiromyces* sp. 2, *C. meandriformis*, *C. venosus*, and *C. magnusii* within the genus *Choiromyces*. *C. cookei*, *C. ellipsosporus*. Among them, *C. venosus* is commonly considered conspecific with *C. meandriformis* [4,8], and we unified the above viewpoint through the phylogenetic analysis results of this study (Figure 3). Among the above species, the *Choiromyces* Chinese specimens formed an independent clade (Figure 3) with high Bayesian posterior probability and were easily distinguishable from other European and American species. Meanwhile, DNA analyses revealed less than 95% ITS sequence similarity between *C. sichuanensis*, *C. cerebriformis*, and *C. helanshanensis*. Most species of fungi show 1–3% intraspecific and >3% interspecific distance between ITS sequences [31]. Thus, the molecular results confirmed that *C. sichuanensis* was well distinguished from *C. cerebriformis* and *C. helanshanensis*. Moreover, morphologically, ascosporal ornamentation in *C. sichuanensis* had longer blunt spine than that in two other Chinese *Choiromyces* species. Therefore, these morphological and molecular results provided strong support for C. *sichuanensis* as a new species (100% and 98% PP). In China, little is known about hypogeous fungi other than the genus *Tuber*. However, the abundance and diversity of *Tuber* species in southwest China indicate a

potential biodiversity of other hypogeous fungi in this region. In this study, we confirmed an independent species within the genus *Choiromyces* and provided fundamental data about the diversity of species for future studies.

The ECM between C. *sichuanensis* and *P. armandii* was well developed, while no such ECM formation was observed on *Picea* roots. Both morphological descriptions and molecular identification fully confirmed the real identities of mycorrhization between *C. sichuanensis* and *P. armandii* as the ITS sequences of the synthesized mycorrhizae were identical to the original ascomata used for inoculation. Although C. *sichuanensis* was sold as Chinese white truffle, the ectomycorrhizal morphological characteristics are different with *Tuber panzhihuanense* which has typical needle-shaped cystidia and pseudoparenchymatous mantles [32]. This result is the first report on the synthesized mycorrhizae between pine plants with a commercial pig truffle of *C. sichuanensis.* Both morphological and molecular evidence in this study could provide reliable references for future studies of other *Choiromyces* species. In contrast, we had not observed ECM formation either from *Picea likiangensis* or *Pi. crassifolia*, although *C. sichuanensis* samples might come from a *Picea* dominated forest, which has to be confirmed in nature. Mycorrhizal synthesis in this study was carried out in September 2021 to March 2022 in greenhouse with natural light and a minimal midday temperature of 10 °C in winter months. Low soil temperatures have been shown to inhibit shoot, leaf, and root growth and water uptake [33–35]. Lahti et al. [36] studied the effects of soil temperature on Norway spruce seedlings' root growth. The results showed that the root growth of Norway spruce was significantly lower at 9 °C compared with the other treatments. In our study, a kind of winter dormancy of *Picea* might inhibit root growth, and then affect the formation of mycorrhiza. Therefore, further studies are required to better understand the relationships between *C. sichuanensis* and *Picea* or other ecotomycorrhizal tree species. The expected results could provide management strategies for developing mushroom cultivation under planted and natural forests.

**Author Contributions:** Conceptualization, F.Y.; methodology, R.W. and S.W.; software, R.W. and S.W.; validation, R.W., S.W. and F.Y; formal analysis, R.W. and S.W.; investigation, S.W.; resources, J.Y.; data curation, R.W. and S.W.; writing—original draft preparation, R.W. and S.W.; writing—review and editing, F.Y.; visualization, R.W. and S.W.; supervision, F.Y.; project administration, S.W.; funding acquisition, S.W. All authors have read and agreed to the published version of the manuscript.

**Funding:** This work is supported by The National Natural Science Foundation of China (for Shanping Wan; grant number: 32060008) and Basic Research Program of Yunnan (for Shanping Wan; grant number: 202201AT070268).

**Institutional Review Board Statement:** Not applicable.

**Data Availability Statement:** Not applicable.

**Acknowledgments:** We are grateful to Xinhua He for constructive comments that improved the quality of the manuscript.

**Conflicts of Interest:** The authors declare that the research was conducted in the absence of any commercial or financial relationships that could be construed as potential conflict of interest.

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
