# Peer review of "Choiromyces sichuanensis sp. nov., a New Species from Southwest China, and Its Mycorrhizal Synthesis with Three Native Conifers"

_diversity, doi:10.3390/d14100837_

Round 1

Reviewer 1 Report

Manuscript “Choiromyces sichuanensis sp. nov., a new edible pig truffle species from Southwest China, and its mycorrhizal synthesis with native trees” by Wang, Wan, Yang and Yu is a comprehensive report of a newly identified species from the genus Choiromyces from China and an attempt to establish ectomycorrhiza of this new species with several conifer species. The manuscript is in general well written, yet it would need to be improved in several aspects to make it suitable for publishing in Diversity journal, as follows:

Title:

In title I would_

-        omit the words “edible pig truffle”, as the manuscript does not give any proof of its (non)edibility (except being sold at the market) and this question is not addressed in the manuscript at all

-        suggest keeping just the Latin name of the species and no English name of the genus, especially since you do not propose an English name for the new species. English name of the genus, e.g. pig truffle is ok to be used as a keyword.

-        Rephrase “with native trees” into more narrow term, that would fit better the content of your paper (for example “three native conifers”)

Abstract:

L14 term “white truffle” is reserved for Tuber magnatum, thus its use here is not in place. In case this name is used in market, then that should be mentioned in discussion, while in abstract I see it as misleading

L15 molecular phylogenetic -> use one or another term, no need to use both for same analysis

L16 “The new described species increases the current number of Choiromyces species to three in China.” – describing new species is not a competition, and describing species is a global scientific approach; rephrase the sentence by omitting “increasing the number” and “three in China”

L17 “mycorrhization synthesis was attempted…”

L21 using term “fundamental” is a bit of an exaggeration. Describing new species and mycorrhization are not revolutionary achievements nowadays

L22 omit term “artificial”, either leave just “mycorrhization” or “mycorrhization in greenhouse”.

L23-24 Mycorrhization and cultivation of edible fungi is a common practice, thus saying “would develop an effective pathway to…” is an overstatement. Rephrase the sentence to “…contributing/broadening the set of species for a successful mycorrhization in greenhouse conditions and potential outplanting for cultivation purposes.”

Introduction

L31-24 omit the data from indexfungorum, as this site list known names of species and does not really reflect the realistic number of species in a given taxonomic units (genus etc.). Although the reclassification to other genera is not relevant to this manuscript. Stay focused on the valid species and existing genus diversity.

L43 “Choiromyces species has been” -> “Choiromyces venosus has been…” as stated in original publication.

L51-53 and L60-66 – I suggest moving objectives of this manuscript to the end of the introduction section as separate paragraph and not “hiding” them in several paragraphs

L66-68 This sentence is obsolete. Rephrased as suggested above, it could be used in conclusions section.

Materials and methods:

In general, I miss any information on statistical tests performed, I guess at least average values, span of values and standard errors were performed for measured parameter if ascocarps, spores, asci,… (see Taxonomic part of results). Please also specify how many ascocarps (from presumably different locations) were analyzed with used approaches (e.g. for light microscopy, for SEM and for ITS sequencing).

L75 please specify “examined”. Currently readers cannot repeat your approach as it is not clear which examinations/tests were you performing under the microscope.

L76-77 Your micrographies how several other structures besides spores. Please specify which structures you attempted to record/to measure (spores, asci, spines,…); see also general comment above

L77-80 Same as in VIS microscopy, also specify which structures you attempted to analyze using SEM

L80 “The specimen has been” this reads as only one specimen has been deposited. Why only one if (at least for molecular data) you show four distinct collections. Please rephrase accordingly to either “The specimen has been deposited...” or “Specimens were deposited…”

L80-81 I miss any information on reference collection(s) of ectomycorrhiza. Do you keep a reference collection? If yes, under which code, and if not, explain why not.

L83 As mentioned below, include in this section also molecular identification of ectomycorrhiza.

L93-95 The sentence “Complete ITS sequences data for 55 taxa were obtained, including 3 new collections and other sequences downloaded from the GenBank database according to previous studies (Table 1).” is misleading. Start with your materials, namely “We have obtained 3 sequences from ascocarps, 1 from ectomycorrhiza, the rest of 51 sequences used for phylogenetic analysis (Table 1) were retrieved from the GenBank (https://www.ncbi.nlm.nih.gov/genbank/).”.

L95 Justify reason(s) why Dingleya was selected as an outgroup (at least cite paper you’ve based your decision upon).

L95 spp.-> sp.

L96 Instead to Fig.1 (for the selected outgroup) please refer to Table 1.

L97 Clarify better how (and how much) of manual editing you’ve done. Keep in mind that described methods have to be repeatable, and as currently stated, no one except you could repeat your analysis

L100 Move Figure 1 to the appropriate position in text.

L107—108 At least for used seeds obtained from the Germplast Bank provide accession numbers

L111-112 rephrase the sentence to “Seeds of all three species were germinated on a sterilized mixture of perlite and peat (1:1, V:V).”.

L112 Here and throughout the text I suggest using term “mycorrhization”

L112 Substrate….was composed of… (not “made”!)

L113 Specify how ascomata were dried (air / forced air (and at which temperature) / freeze-dried / …). Specify also how many ascomata were used (one or mixture of several from same/different origin)

L117 Clarify how you have determined the number of spores per seedling

L117-118 “Ten milliliters of spore slurry were distributed with a pipette around” – in previous sentence you already specified the number of spores per seedling. There is no need to re-specify this in volume of slurry applied. Shorten sentence to “Spore slurry was applied in upper third of the seedlings root system.”

L119 Rephrase “Non-C. sichuanensis inoculated control seedlings” to “Negative control seedlings…”. Please specify how many seedlings (per each species!) were used as negative controls (currently this information is only hidden in L124-125.

L121 “natural light” -> “ambient light conditions”; please add any information regarding climate conditions in the greenhouse.

L122-123 I would omit/rephrase the sentence “The containers with or without C. sichuanensis inoculation were grown in groups, rather than a randomized arrangement.” as you primarily aimed to establish ectomycorrhiza and the distribution of seedlings is not relevant, unless of course you would have problems with cross contaminations.

L124 pine/Pinus – do not mix English and Latin terms. Use one or another; I would prefer Latin (throughout the manuscript).

L125-128. This explanation is showing your cultivation in bad light. There is not need to discuss this (definitely not in Methods, potentially in Discussion in case you would find it relevant and detrimental for your results).

L128-129 As substrate is composed of peat, preventing drying it out is crucial – I would suggest to add some information on how pots were watered and how much water (per seedling and ped watering cycle) was used.

L131-134. This sentence is too long. Make at least two, one on ho root systems were treated and cleaned and one in characterization of ectomycorrhiza

L139-146 I suggest implementing the molecular identification of ectomycorrhiza into paragraph 2.2. That would shorten the manuscript and keep comparable analyses in same paragraph.

L140 Please explain term “pooled mycorrhizal tips” (pooled from same plant, different plants of the same host,….).

L139-146 Clarify how many ectomycorrhiza samples (not root tips!) per plant×fungus were analyzed. Did you observe any “non-typical” ectomycorrhiza that was included in molecular analysis (potential contamination!!). Did you also extract DNA from roots of negative controls and roots from Picea (e.g. where you observed no formed ectomycorrhiza)? If no, please explain why; if yes, please state how many samples and how you have decided which roots to analyse

Results

General comment: Change order of “Remarks” following the principle to firstly combine the new species with other species from the same genus (e.g. Choiromyces) and only after broaden the comparison to other genera (e.g. Tuber)

L151 Stating that type collection was recorded from under Picea (asperata) and later in text claiming this species cannot form ectomycorrhiza with two other Picea species provoke two questions 1.) if this Choiromyces does not form ectomycorrhiza with Picea, how could then be found in (pure?) Picea stand? and 2.) If the record of the type specimen was from under Picea asperata forest, why you did not use this particular species for mycorrhization? Please try to reply these two questions in Introduction and/or Discussion.

L151 Please add author’s name for Picea asperata.

L157-158 Use same number of decimals in “0.7–10 μm” (depending on an accuracy of your measurements)

L170 the statement “the true “white” truffle of Tuber panzhihuanense” is misleading. In general, using terms “white” and “black” truffles has no taxonomic value and should be omitted in scientific literature. Rephrase this sentence to “This species shares similar characteristics with Tuber panzhihuanense, namely size of ascocarps, colour, an irregularly knobbed ascomata, and whitish gleba with sinuous veins when young.”

L174, L176 Omit “Chinese” as species you compare it with may not be only distributed in China

L176 Add missing spaces (“was <3 μm” -> “was < 3 μm”) and use same number of decimals (“2-2.5 -> 2.0-2.5”)

L189 “specimens and ECM” -> “(Table 1)”

L193-194 “one Choiromyces-like mycorrhiza from this study”. Why “Choiromyces-like”? After all you probed to by Choiromyces ECM! ; simplify this sentence.

L195 Delete “Chinese”

L197 The time and period of the inoculation should be already specified in Methodology. Please add to the period also climatic belt information and when the growth season if used tree species is expected. For example, in continental Europe most of Picea species remain dormant due to the “winter period”, thus in this period mycorrhization is not successful at all. Could that be a reason you have failed with mycorrhization of Picea? At least Picea crassifolia grows in areas with harsh winter times and such winter dormancy would be expected. Please elaborate on this in Discussion.

L198-199 I miss any quantification of mycorrhization. What was % of mycorrhized roots in Pinus armandii?

L199-L207 Please describe ECM more in details. Current description is insufficient and not convincible. Some information, not described here can easily be seen from Figure 2 and 3. Be precise, describe ECM in details and follow terminology of Agerer (for example Agerer 1991 and/or Agerer 1987-2013 (Colour atlas of Ectomycorrhizae) and/or an online terminology used in deemy.de).

L200-201 Mention how you have assessed colours. I would suggest using one if the official colour charts (and colour codes therein), for example Munsells chart or Pantone,…

Discussion

In general, I would suggest authors to structure discussion in a way to clearly follow your objectives. Start with the more significant outcome(s) of your study, followed by less, or negative results. Please make sure you discuss your outcomes and not only repeat your results (e.g. L228-232). I miss any references towards which you discuss your outcomes with. After all there were published several analyses of Choiromyces phylogeny. Include those, and again, do not limit your discussion solely to China; such approach in scientifically not correct.

L225-227 From the scientific point of view it is less relevant how many species are in China. Either refer to ecological unit, climate, vegetation, or omit this sentence. As mentioned earlier, this is not a race and indexing species. Consider discussing your work in a global scope. In this sense (as above), delete “Chinese” in L228, L229, L232.

L238-247. I miss at least a brief discussion on comparing ECM characteristics of C. sichuanensis with other (morphologically similar ECMs). Describing new ECM without comparison to main ECM publications and databases (Agerer 1987-2013 (Colour atlas of Ectomycorrhizae), Rambold and Agerer 2014 (deemy.de) or several American publications) is not comprehensive and insufficient for publishing. Please make sure to include such critical comparison in text.

The last paragraph refers to a functional effect of mycorrhization, which was not a subject of your paper. Please stick strictly to objectives of your manuscript.

Tables

Table 1

-        Add authorities to all species names

-        Table caption should include all parameters, you present. Use same terminology (Voucher number = “source of specimen” and delete “used in this study” as this is understandable from the content of the Table. Add also the range of taxa listed (e.g. List of collections and sequences from the genus Choiromyces, and sequences used as an outgroup. Species name, voucher,….are given”

-        Instead of “Voucher number” use “Voucher”

-        In first line of the table write Choiromyces in full

-        The correct way of giving name of ectomycorrhiza is combining fungal partner × plant partner name (including authorities). For your ectomycorrhiza the full name would be “Choiromyces sichuanensis S. P. Wan, R. Wang & F. Q. Yu × Pinus armandii Franch.”

Figures:

Figure 1

-        I would suggest using only black and white for this figure. Type new species (and combination in ECM) with bold text.

-        Make sure same typography is used

-        Why marking “Clade 1” is no other (existing) clades are marked. I would delineate at least three. In case you decide to keep “clades” marked, please refer to them in text and explain in results their relations

Figure 3

-        In caption, please describe better which structure(s) are shown on each of the subfigures. Fig. 3e for example is confusing as it seems to show a rhizomorph, while their presence is not mentioned in text and are not visible in other figures (Fig3b,c,d,g,h)

-        Figs 3g,h and I basically show same thing. Use only one and mark all structures therein.

-        Fig 3f Add mantle type (letter A-Q) following Agerer 1991 (or other references mentioned above)

Figure 4

-        A closer look at the roots might be beneficial (on Fig 4d there might be some root tips mycorrhized)

-        Make sure same typography is used in caption.

Reviewer 2 Report

The manuscript is interesting and clearly written A new species of Chyromyces from China  was described and its mycorrhizas were obtained in greenhouse.  To my knowledge that is the first time that the ectomycorrhizas of a Choiromyces sp. are described.

However there are several points which need to be addressed. The evidence of new species should be  supported by a multi-locus DNA analyses. What is the ITS similaritity between this  new species and  the other Chinese Choiromyces spp.?

Although it is not mandatory a short diagnosis should be added at the species description (see https://doi.org/10.1186/s43008-021-00063-199).

There are also several minor points which need to be addressed:

Abstracts lines 13-25 : all the latin names should be in italic

Line 30 it can not be described in 1843 and publishes before in 1831!

In index fungorum only six species of have been categorized. The new species C. cerebriformis is published butit seems not recognized and the current name of C. ellipsosporus is Hydnotryopsis setchellii Gilkey.

Line 43That is not true.  It is considered a delicacy only in same countries of Northern Europe like Norway . In Sothern Europe it  is even considered slightly toxic.

Line 49 please indicate the species of white truffe (Tuber panzhihuanense? And others?)

Line 65 P. not Pi, please check all the manuscript.

Line 81 please write in the herbarium of Yunnan Agricultural University….Does have the herbarium an official name?

Line 152 I prefer if you include a short diagnosis

Line 156 mm? Should be µm!

Lines 167-172 I personally think that you have to describe here only the differences between the different species of Choiromyces. The similarity with true truffle could be part of the discussion as well as the possible frouds in the market.

Line 198 What about the mycorrhization degree?

Line 243, do you have the evidence of the description of the mycorrhizas of other Choiromyces species? In the case please compare. You can also compare the Choiromyces mycorrhizas with those of the sister genus Tuber.

Round 2

Reviewer 2 Report

The manuscript is considerably improved after revision. Although a multi-locus DNA analysis was not applied, the similarity of ITS sequences is low and it can support a new species.

There are only a few minor points, which still needs to be addressed before publication:

Lines 63-65: I don't like this sentence

In the table 1: "C. Meandriformis" should be "C. meandriformis"

Line 208: Please modify as: "Hartig net palmetto type with a single hyphal row"

Figure 3: In my opinion, it could be omitted.
